# Geometrical Analysis of the Inland Topography to Assess the Likely Response of Wave-Dominated Coastline to Sea Level: Application to Great Britain

**Andres Payo ***[ID]**, Chris Williams** [ID]**, Rowan Vernon, Andrew G. Hulbert, Kathryn A. Lee and Jonathan R. Lee**

British Geological Survey, Nicker Hill, Keyworth, Nottingham NG12 5GG, UK; chrwil@bgs.ac.uk (C.W.); rowver@bgs.ac.uk (R.V.); agh@bgs.ac.uk (A.G.H.); kbo@bgs.ac.uk (K.A.L.); jrlee@bgs.ac.uk (J.R.L.)
* Correspondence: agarcia@bgs.ac.uk; Tel.: +44-0115-936-3103

**Abstract:** The need for quantitative assessments at a large spatial scale ($10^3$ km) and over time horizons of the order $10^1$ to $10^2$ years have been reinforced by the 2019 Special Report on the Ocean and Cryosphere in a Changing Climate, which concluded that adaptation to a sea-level rise will be needed no matter what emission scenario is followed. Here, we used a simple geometrical analysis of the backshore topography to assess the likely response of any wave-dominated coastline to a sea-level rise, and we applied it along the entire Great Britain (GB) coastline, which is ca. 17,820 km long. We illustrated how the backshore geometry can be linked to the shoreline response (rate of change and net response: erosion or accretion) to a sea-level rise by using a generalized shoreline Exner equation, which includes the effect of the backshore slope and differences in sediment fractions within the nearshore. To apply this to the whole of GB, we developed an automated delineation approach to extract the main geometrical attributes. Our analysis suggests that 71% of the coast of GB is best described as gentle coast, including estuarine coastline or open coasts where back-barrier beaches can form. The remaining 29% is best described as cliff-type coastlines, for which the majority (57%) of the backshore slope values are negative, suggesting that a non-equilibrium trajectory will most likely be followed as a response to a rise in sea level. For the remaining 43% of the cliffed coast, we have provided regional statistics showing where the potential sinks and sources of sediment are likely to be.

**Keywords:** erosion; sea-level rise; nearshore

## 1. Introduction

One of the most significant challenges facing coastal geomorphology and engineering today is the need to improve our ability to make quantitative predictions of morphological change at a scale that is relevant to longer-term strategic coastal management [1]. Following [2], this scale is herein referred to as the mesoscale, and is characterised by time horizons of the order of $10^1$ to $10^2$ years. The need for these quantitative assessments at the mesoscale has been reinforced by the 2019 Special Report on the Ocean and Cryosphere in a Changing Climate [3], which concluded that adaptation to a sea-level rise ($R$) will be needed no matter what emission scenario is followed. As a response to rising sea levels, the location of the shoreline is anticipated to change, with the magnitude and direction of the change (transgression or progradation) being a continuum between two extreme behaviours: passive inundation and morphodynamic evolution. Passive inundation occurs when the land surface is static during transgression and therefore shoreline retreat follows the slope, $S_0$, of the

backshore topography. Morphodynamic evolution is usually accompanied by erosion and deposition, which drive morphologic changes that impact future retreat.

Which of these two extreme behaviours is more likely to occur can be understood from knowledge of the backshore slope and the different instances of the Bruun rule [4]. The Bruun rule is the most common model used for practical predictions of shoreline retreat, particularly in the context of global warming. While simple to use, the Bruun rule makes simplifying assumptions, which are frequently misunderstood or ignored in practice [4]. The Bruun rule requires the following three assumptions: (1) the shoreface, beach, and substrate must have a homogeneous composition; (2) the shoreface and beach must maintain a fixed equilibrium profile; and (3) this profile must be closed with respect to external sources and sinks of sediment. Different instances (classical, back-barrier and cliff Bruun rules) can be obtained [4] by applying the conservation of mass, under the Bruun rule assumptions, over different control volumes to produce a relationship where the landward rate of movement of the shoreline, $\varepsilon$ (distance/time), is a linear function of the sea-level rise itself (R, distance/time):

$$\varepsilon = R/\overline{S} \tag{1}$$

where $\overline{S}$ represent the effective slope controlling the sea-level rise response. Which geometric boundaries accurately represent the effective slope, $\overline{S}$, has been an issue of significant debate. Although some applications suggest that the effective slope should be the shoreface slope, $S_s$, or, for low-sloped coasts, some other modification accounting for barrier geometry [5], recent analytic research suggests that long-term equilibrium requires that the coastal recession rate follow the regional upland slope [6–8]. When the conservation of mass is applied to the shoreface, as shown in Figure 1a, where $\overline{S} = S_S$ under the Bruun essential assumptions provides the classic Bruun rule, the shoreface profile translates upwards following a sea-level rise and landwards following the backshore slope. However, on gentle coasts ($S_0 < S_s$) the classic Bruun rule fails to account for back-barrier deposition. In this case, extending the Bruun profile to include the barrier and back-barrier (Figure 1b) reduces the average profile slope $\overline{S}$, giving a "barrier Bruun rule" where $\overline{S} < S_s$. For a barrier of sufficient length, $\overline{S} = S_0$, and the barrier Bruun rule reduces to passive inundation, here accomplished by barrier rollover. Similarly, on steep coasts ($S_0 > S_s$), extending the Bruun profile to include a cliff face (Figure 1c), accounting for cliff erosion, increases the average profile slope $\overline{S}$, giving a "cliff Bruun rule" where $\overline{S} > S_s$, and for a tall-enough cliff, the cliff Bruun rule again reduces to passive inundation. Equation (1) shows how the backshore slope plays an important role in determining the coastal response to a sea-level rise, but caution is needed when applying the Bruun rule, as its assumptions are not always met in reality and have to be applied with care. Much of the controversy surrounding the Bruun rule assumptions concerns the validity of the closed equilibrium profile concept (Assumptions 2 and 3), but [7] argued that the least robust assumption is the assumption that negligible sediment flux across the shoreline is required to fully close the profile. The violation of the negligible sediment flux across the shoreline assumption leads to physically unreasonable long-term predictions for the Bruun shoreline implying cliff erosion or back-barrier deposition when applied to coasts steep or gentle relative to the shoreface slope. To overcome this limitation, a generalized Bruun rule that includes the landward sources and sinks of sediment was proposed by [7].

Here, we propose an innovative approach to assess the potential shoreline response to rising sea levels based only on the geometrical analysis of backshore topographical profiles, and we applied it to the whole of the Great Britain-and-isles coastline (hereinafter referred as GB). First, we show how Equation (1) was generalized by [7] to include the sources and sinks of sediment for steep and gentle backshore slopes to produce a generalized cliff and back-barrier Bruun rule, respectively. To be able to obtain an analytical solution of the generalized Bruun rule, [7] assumed that backshore relief can be characterized by a constant slope. To characterize the backshore topography of the length of the GB coastline (ca. 17,820 km), we first extracted the topographical profiles using an improved version of the automatic delineation approach proposed by [9] at 50 m intervals from the BlueSky 2 m Digital Terrain Model (DTM). We then extracted not only the backshore slope but also a set of geometrical attributes

to assess how much the backshore relief deviated from the constant slope assumption. To facilitate the interpretation of the results, we clustered the profiles via a combination of statistical clustering and expert elucidation. This approach enabled (1) the classification of the types of coastline morphology that occur around the GB coastline; (2) an estimation of the main contributions to the generalized sediment conservation equation for each coastline type; and (3) the attribution of the entire Great Britain coastline with these coastline types.

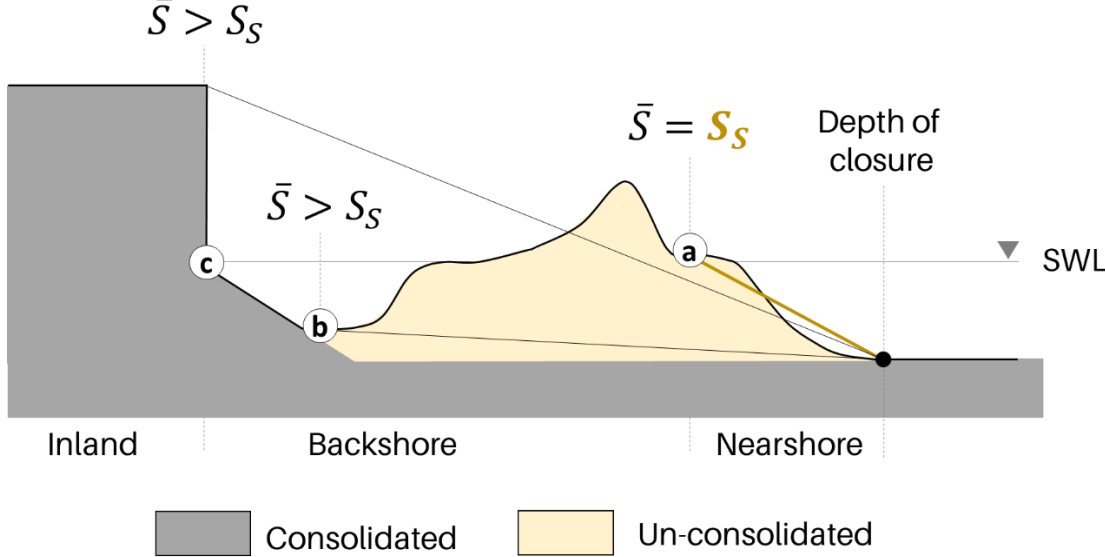

**Figure 1.** Applications of the essential Bruun rule to different control volumes produce different instantiations. (**a**) Application to a shoreface and beach gives the classic Bruun rule. (**b**) Application to a beach and shoreface backed by a barrier island gives a barrier Bruun rule. (**c**) Application to a beach and shoreface backed by a cliff gives a cliff Bruun rule. In each case, the implied transgression slope (black dashed line) equals the average profile slope $\bar{S}$, but only in the classic Bruun rule is this equal to the nearshore slope $S_s$ (gold solid line). (Adapted by authors from [7]).

The paper provides a detailed overview of the methodology and data employed (Section 2) before outlining the main results of the study (Section 3). The significance of the results in relation to producing a typology of the Great Britain coastline as well as an assessment of the likely response to a sea-level rise is discussed in Section 4.

## 2. Materials and Methods

### 2.1. Generalized Sediment Conservation Equation for Steep and Gentle Backshore Slopes

From the general analysis of the conservation of sediment over a cross-shore profile $z = \eta[x]$ at a fixed alongshore position y, using a shoreface control volume $s \leq x \leq u$ (Figure 2) [7] derived a generalized shoreline transgression for any coastal profile as:

$$\underbrace{c_0 H \frac{ds}{dt}}_{shoreline\ migration} = \underbrace{q_{x,s} - q_{x,u}}_{cross-shore\ flux\ divergence} - \underbrace{\partial_y Q_y}_{alongshore\ flux\ divergence} - \underbrace{\bar{c} L R'}_{accomodation} \qquad (2)$$

Each variable in Equation (2) is described in Table 1, and each term can be understood as a cross-sectional area of sediment produced (or consumed) per unit time. The left-hand side is sediment produced by coastal erosion due to shoreline retreat (for $ds/dt < 0$). The right-hand terms represent sediment accumulation due to net cross- and along-shore sediment flux divergences, and sediment consumption required to keep pace with a sea-level rise at an effective rate $R'$. Table 1 summarizes the variables shown in Figure 2. Note that Equation (2) requires $\eta_s - z_{sea} = const$ but is insensitive to the

particular datum used to define the shoreline $s$. The authors of [6] provided a full derivation of the shoreline Equation (2) and discussed its significance generally.

**Table 1.** Shoreline sediment conservation equation parameters [a].

| Symbol | Variable | Units |
|---|---|---|
| $x$ | cross-shore coordinate | m |
| $y$ | alongshore coordinate | m |
| $z$ | vertical coordinate | m |
| $t$ | time | s |
| $\eta(x)$ | surface elevation | m |
| $s$ | cross-shore shoreline position | m |
| $u$ | cross-shore profile toe elevation | m |
| $\eta_s$ | shoreline elevation | m |
| $\eta_u$ | profile toe elevation | m |
| $q_x$ | cross-shore sediment flux | $m^2/s$ |
| $Q_y$ | alongshore sediment discharge | $m^3/s$ |
| $\Delta q$ | net flux difference | $m^2/s$ |
| $L$ | profile length | m |
| $H$ | net profile relief | m |
| $S$ | average profile slope | m/m |
| $\alpha$ | profile shape factor | $m^2/m^2$ |
| $R$ | relative sea-level rise rate | m/s |
| $R'$ | effective sea-level rise rate | m/s |
| $\bar{c}$ | average sand concentration | $m^3/m^3$ |
| $c_0$ | shoreline sand concentration | $m^3/m^3$ |

[a] as used in Figure 2.

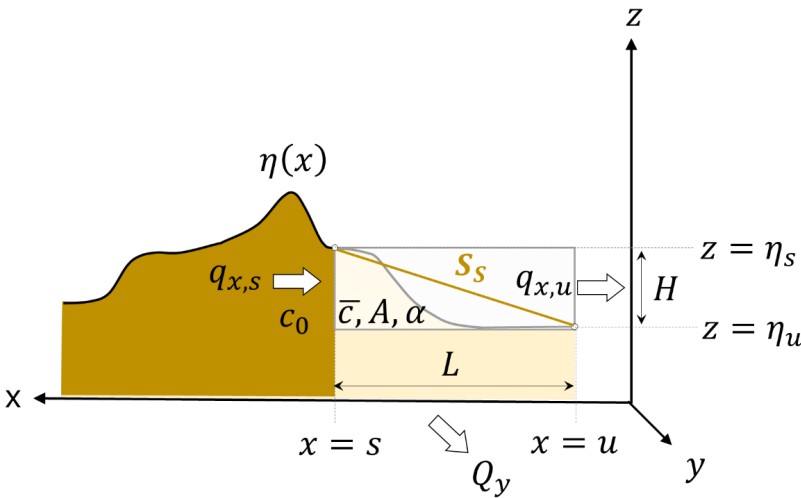

**Figure 2.** The shoreline response to a sea-level rise for both passive inundation and morphodynamic evolution is best understood from a general analysis of the conservation of sediments over a cross-shore profile $z = \eta[x]$ at a fixed alongshore position $y$, using a shoreface control volume $s \le x \le u$. The sediment conservation Equation (1) assumes the control volume contains a relatively homogeneous deposit with an average sediment concentration $\bar{c}$, but allows the possibility of a compositional change at the shoreline $s$, where the sediment concentration $c_0$ may differ from $c$. In general, $c$ may represent bulk sediment or sediment of a particular size range. Symbols explained in Table 1. (Figure adapted by authors from [7]).

The effective sea-level rise rate $R'$ includes accommodation due to both a relative sea-level rise and changes in profile geometry.

$$\underbrace{R'}_{effective\ sea\ level} = \underbrace{R}_{relative\ sea\ level} + \underbrace{(2\alpha - 1)\overline{S}\frac{dL}{dt}}_{length} + \underbrace{(\alpha - 1)L\frac{d\overline{S}}{dt}}_{slope} + \underbrace{H\frac{d\alpha}{dt}}_{shape} \tag{3}$$

where the bulk profile geometry is described completely by the length $L$, relief $H$, average slope $\overline{S}$, and average shape a (Figure 2). These variables are related by

$$\overline{S} = \frac{H}{L}, \quad \alpha = \frac{A}{LH} \tag{4}$$

Note that the order-one shape factor $\alpha$ depends only on the fractional sediment fill of the control volume bounding box. For smooth monotonic profiles, $\alpha$ also describes the profile curvature: $\alpha = 1/2$ for a linear profile, while $\alpha < 1/2$ for a concave profile, and $\alpha > 1/2$ for a convex profile.

The mathematical expression for the cliff shoreline evolution model shown in Figure 3 is obtained by applying the shoreline Exner Equation (2) to the vertical cliff (i.e., zero length) and the nearshore region and then combining them into a net cliff-nearshore mass balance equation resulting in

$$\frac{ds}{dt} = -\left(\frac{c_s}{c_0}\right)\frac{L_s R}{H_s + H_c} = -\left(\frac{c_s}{c_0}\right)\frac{R}{\overline{S}}, \tag{5}$$

where $\overline{S}$ is the average slope of the combined cliff and nearshore profile (Figure 3). This simplified model assumes cliff erosion ($\frac{ds}{dt} < 0$) produces a seaward flux of sediment ($q_s > 0$) in concert with shoreline retreat and produces sediment that is instantly available to feed nearshore aggradation. For a rocky coast, cliff retreat occurs only after shoreline retreat undercuts the cliff, and produces rock fragments, which must be weathered before sand becomes available to the nearshore. In the special case where cliff and nearshore deposits are homogeneous, $c_0 = c_s$, Equation (5) reduces to the cliff Bruun rule of Equation (1) and Figure 1c. However, the slope $\overline{S}$ will vary as the cliff relief $H_c[t]$ changes through time.

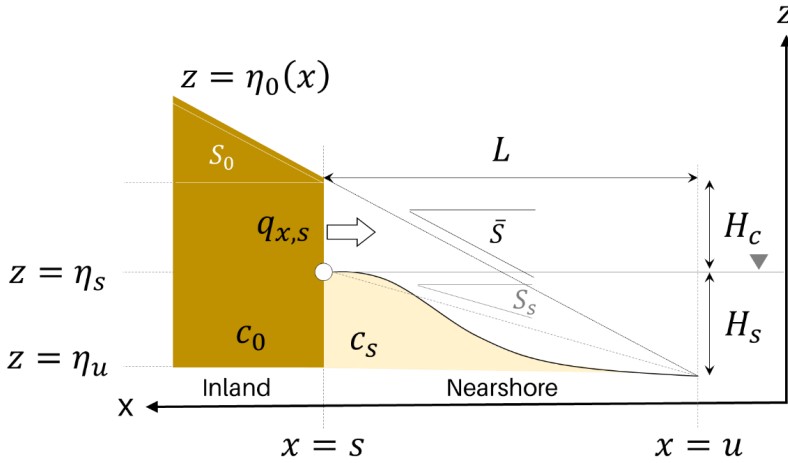

**Figure 3.** Idealized geometry and mass balance on a steep coast. Nearshore geometry is fixed but cliff height $H_c$ can vary in time. Shoreline $s$ located at cliff toe. Beach top elevation $\eta_s$ rises with sea level $z_{sea}$, maintaining a fixed beach relief. Cliff erosion produces sediment flux $q_s$, which fuels nearshore aggradation. Sand fraction $c_0$ of cliff deposits may differ from nearshore sand fraction $c_s$. (Figure adapted by authors from [7]).

At any time $t$, the cliff relief $H_c[t]$ is the difference in elevation between the backshore topography $\eta_0[s]$ and the shoreline elevation $\eta_s[s]$. If we define the origin of our (x,z) coordinate system to be the shoreline position $(s, \eta_s)$ at time $t = 0$, then from the geometry of Figure 3, the cliff relief at time $t$ is

$$H_c[t] = H_{c,0} - S_0 s - Rt,\tag{6}$$

where $H_{c,0}$ is the cliff relief at time $t = 0$. Differentiating Equation (6) with respect to time gives

$$S_0 \frac{ds}{dt} = -\frac{dH_c}{dt} - R,\tag{7}$$

which can be combined with Equation (5) to eliminate $ds/dt$, giving an ordinary differential equation (ODE) for the time evolution of the cliff relief $H_c[t]$

$$\frac{dH_c}{dt} = R\left(\frac{H_{c,\infty} - H_c[t]}{H_s - H_c[t]}\right),\tag{8a}$$

$$\begin{aligned} H_{c,\infty} &= \left(\frac{c_s}{c_0}\right) S_0 L_s - H_s, \\ H_c[0] &= H_{c,0}, \end{aligned}\tag{8b}$$

where $H_{c,\infty}$ is the equilibrium cliff relief reached as $t$ becomes large. Equations (8a)–(8b) constitute [7]'s model for shoreline retreat on steep coasts. On steep coasts, shoreline retreat may obey the classic Bruun model in the short term, but in the long run, shoreline retreat will always converge to the passive inundation model. A first-order estimate for the equilibrium transition is the time for the sea level to traverse the effective zone of erosion [8], which is determined by the tidal range in conjunction with wave variability. For example, with a 1 m and 10 m elevation of the zone of erosion, under a sea-level rise rate of 10 mm/year, it will take 100 to 1000 years to fully traverse this zone and reach equilibrium. To determine shoreline behaviour over intermediate timescales, we must solve Equation (8a) for a time-varying cliff relief $H_c[t]$, from which we can compute the shoreline trajectory $s[t]$ using Equation (6). The solution to Equation (8a,b) was given by [7] as

$$H_c[t] = H_{c,\infty} + (H_s + H_{c,\infty}) \times W\left[\left(\left(\frac{H_{c,0} - H_{c,\infty}}{H_s + H_{c,\infty}}\right) exp\left[\frac{H_{c,0} - H_{c,\infty}}{H_s + H_{c,\infty}}\right]\right) exp\left[-\frac{Rt}{H_s + H_{c,\infty}}\right]\right],\tag{9}$$

where W[] is Lambert's W function [10].

The mathematical expression for the back-barrier shoreline evolution model shown in Figure 4 is obtained by applying the shoreline Exner Equation (2) to the shoreface, overwash zone and back-barrier regions and then combining them into a net backshore-barrier-island and nearshore mass balance equation resulting in

$$\frac{ds}{dt} = -\left[\frac{L_s + L_w + L_b}{(c_0/c_s)H_s - H_b}\right]R + (1 - 2\alpha_b)\left[\frac{L_b}{(c_0/c_s)H_s - H_b}\right],\tag{10}$$

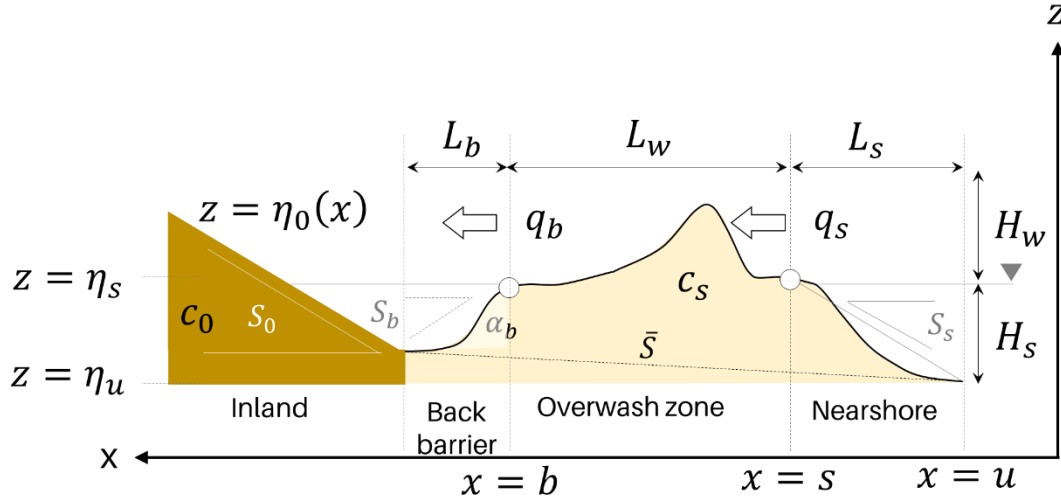

**Figure 4.** Idealized geometry and mass balance on a gentle coast. Barrier island consists of subaqueous shoreface, subaerial island (overwash zone) and subaqueous back-barrier. Island bounded seaward by shoreline $s$ and landward by backshore $b$, both at sea level $z_{sea}$. Shoreface and island have fixed geometry. Back-barrier slope $S_b$ and shape $\alpha_b$ are fixed, but back-barrier length $L_b$ and relief $H_b$ can vary in time. Island maintained by overwash flux $q_s$ across the shoreline. Back-barrier growth fuelled by overwash flux $q_b$ across the backshore. Sand fraction $c_s$ of barrier island deposits may differ from sand fraction $c_0$ of backshore deposits. (Figure adapted by authors from [7]).

At any time $t$, the back-barrier relief $H_b[t]$ is the difference in elevation between the backshore topography $\eta_0[b-L_b]$ and the shoreline elevation $\eta_s[s] = z_{sea}[t]$. If we define the origin of our (x, z) coordinate system to be the shoreline position $(s, \eta_s)$ at time $t = 0$, then from the geometry of Figure 4, the back-barrier relief at time $t$ is

$$H_b[t] = H_{b,0} - \frac{S_0 s + Rt}{1 + S_0/S_b},$$

(11)

where $H_{b,0}$ is the initial back-barrier relief at time $t = 0$. Differentiating Equation (10) with respect to time gives

$$S_0 \frac{ds}{dt} = (1 + S_0/S_b)\frac{dH_b}{dt} - R,$$

(12)

which can be combined with Equation (10) to eliminate $ds/dt$, giving an ordinary differential equation (ODE) for the time evolution of the back-barrier relief $H_b[t]$

$$\frac{dH_b}{dt} = \widetilde{R}\left(\frac{H_{b,\infty} - H_b[t]}{\widetilde{H_s} - H_b[t]}\right),$$

(13a)

$$H_{b,\infty} = \frac{(c_0/c_s)H_s - S_0(L_s+L_w)}{1+S_0/S_b},$$
$$H_b[0] = H_{b,0},$$

(13b)

where $H_{b,\infty}$ is equilibrium back-barrier relief, and the effective shoreface relief $\widetilde{H_s}$ and sea-level rise rate $\widetilde{R}$ are defined by

$$\widetilde{H_s} = \beta(c_0/c_s)H_s, \quad \widetilde{R} = \beta R, \quad \beta = \left(1 + \frac{1-2\alpha_b}{1+S_0/S_b}\right)^{-1},$$

(14)

where $\beta$ is a shape factor accounting for back-barrier curvature. In typical cases, where the curvature is mild ($\alpha_b = 1/2$) and the back-barrier is relatively steep ($S_b \gg S_0$), this shape factor is near one.

Equations (13) and (14) constitute our model for shoreline retreat on gentle coasts. From Equation (13), we can see immediately that as in the steep coast case, in the long-term limit

of an equilibrium back-barrier ($H_b = H_{b,\infty}$), shoreline transgression converges to passive inundation. Similarly, to determine shoreline behavior over intermediate timescales, we must solve Equation (13a) for time-varying back-barrier relief $H_b[t]$ and then compute the shoreline trajectory $s[t]$ using Equation (11). As shown by [7], the solution to Equation (13a–c) is given by

$$H_b[t] = H_{b,\infty} - \left(\widetilde{H_s} - H_{b,\infty}\right) \times W\left[\left[\left(\frac{H_{b,0} - H_{b,\infty}}{\widetilde{H_s} - H_{b,\infty}}\right) exp\left[\frac{H_{b,0} - H_{b,\infty}}{\widetilde{H_s} - H_{b,\infty}}\right]\right] exp\left[-\frac{\widetilde{R}t}{\widetilde{H_s} + H_{b,\infty}}\right]\right], \qquad (15)$$

where W[] is Lambert's W function [10]. In our gentle coast model, the fundamental requirement for barrier island formation is net onshore sediment transport ($q_s < 0$). As shown by [7], for an equilibrium barrier, we can use Equations (2) applied to the shoreface shown in Figure 4 and (12) to express this as

$$q_{s,eq} = -RH_s\left(\frac{c_0}{S_0} - \frac{c_s}{S_s}\right), \quad S_s > \left(\frac{c_s}{c_0}\right)S_0, \qquad (16)$$

Hence, even if $S_s > S_0$, barrier formation may be impossible on a sand-poor substrate ($c_0 < c_s$). On the other hand, onshore sediment transport does not guarantee barrier island formation. An equilibrium back-barrier is only possible if $H_{b,\infty} > 0$, which requires

$$q_{s,eq} > c_s L_w R, \qquad (17)$$

The onshore flux, $q_{s,eq}$, must be sufficient to aggrade the island while leaving some sediment left over for back-barrier aggradation [7]; otherwise, a barrier beach will form rather than a true barrier island. More generally, we can interpret $L_w$ as a maximum overwash distance. Physical limitations on overwash processes will always impose a maximum length, $L_{w,max}$, as a function of both storm and wave climate (e.g., surge heights) and sediment supply limitations, such that

$$L_{w,max} = min\left\{L_{w,max}, -q_{s,eq}/(c_s R)\right\}, \qquad (18)$$

When Equation (17) is satisfied, we get a barrier island with $L_w = L_{w,max}$ and a subaqueous back-barrier, but when Equation (17) is violated, we get a barrier beach with $L_w < L_{w,max}$ and no back-barrier. Combining Equations (16) and (17) shows that a true barrier island will form only if

$$S_s > \left(\frac{c_s}{c_0}\right)\left(1 + \frac{L_w}{L_s}\right), \qquad (19)$$

Equations (16) and (19) delineate a continuum of coastal landforms that can occur on gentle coasts experiencing a sea-level rise. If Equation (16) is violated, we get a cliff. Otherwise we get a barrier beach with a length given by Equation (18), or if Equation (19) is also satisfied, we get a barrier island.

The role of backshore topography in shoreline change due to a sea-level rise can be inferred from Equations (9) and (15). We start by noticing that the two real branches $W_{k=0,1}$ of the Lambert function W have a shape as shown in Figure 5. Any initial state value that falls within the domain of the principal branch, $W_{k=0}$, will evolve from the initial $H_{c,0}$ and $H_{b,0}$ towards the equilibrium $H_{c,\infty}$ and $H_{b,\infty}$, respectively. Any initial state value that falls within the domain of the branch $W_{k=1}$ will evolve towards $-\infty$ and will never reach equilibrium. The slope of the W function at $t = 0$ indicates the rate of change of $H_c[t]$ and $H_b[t]$ over time. From Figure 5, it can be seen that the slope of $W(x)$ is maximal and near vertical at the $x = -1/e$ value (i.e., where the two branches coincide); the slope is approximately unity near $x = 0$ and smaller than unity for $x >> 0$. The initial value of $x$ on $W(x)$ can be obtained from the identity $W(x = ze^z) = z$ of the Lambert $W_{k=0,1}$ branches as

$$z_c = \frac{H_{c,0} - H_{c,\infty}}{H_s + H_{c,\infty}} = \frac{H_{c,0} - \left(\frac{c_s}{c_0}\right)S_0 L_s + S_s L_s}{\left(\frac{c_s}{c_0}\right)S_0 L_s} = \frac{H_{c,0} + S_s L_s}{\left(\frac{c_s}{c_0}\right)S_0 L_s} - 1 = \frac{\overline{S} + S_s}{\left(\frac{c_s}{c_0}\right)S_0} - 1, \quad \overline{S} = H_{c,0}/L_s, \qquad (20)$$

$$z_b = \frac{H_{b,0} - H_{b,\infty}}{\widetilde{H_s} - H_{b,\infty}} = \frac{H_{b,0} - H_{b,\infty}}{\beta(c_0/c_s)H_s - H_{b,\infty}}. \tag{21}$$

where $z_c$ and $z_b$ are the initial values of $z$ in $x = ze^z$ for the cliff and back-barrier models of Equations (9) and (15), respectively, and are functions of both the geometry of the coast and the sediment fraction composition. Equations (20) and (21) provide a simple geometrical relationship between the topography and the sediment fraction ratio for $z_c$ and $z_b$ to be equal to −1, 0 and > 0, which corresponds with the x values $x = -1/e$, $x = 0$ and $x >> 0$, with the maximum, median and minimum characteristic rates of change of W with x. From the expected rates of change of $H_c[t]$ and $H_b[t]$ and Equations (7) and (12), we can also infer if the shoreline position will retreat ($\frac{ds}{dt} < 0$) or advance ($\frac{ds}{dt} > 0$) as the sea-level rises. We first notice that the condition for the shoreline to advance requires $\frac{dH_c}{dt} < 0$ and $\frac{dH_b}{dt} > 0$ for our cliff and back-barrier models, respectively. Hereinafter, we will only consider the main branch $W_{k=0}$ solution, which will continuously evolve towards equilibrium, and therefore, the condition for shoreline advance under a sea-level rise can be expressed as

$$H_{c,0} > H_{c,\infty}; \ H_{b,0} < H_{b,\infty}, \tag{22}$$

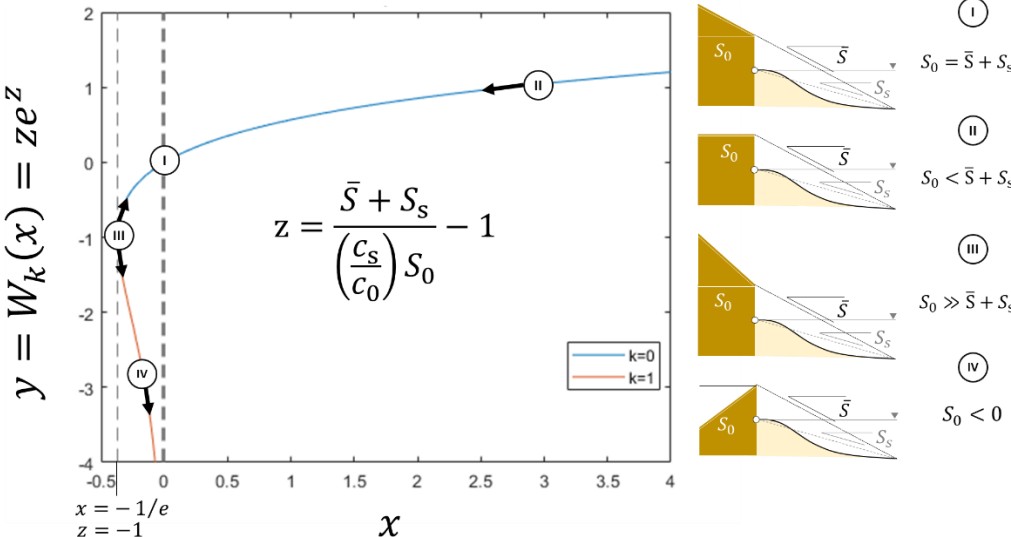

**Figure 5.** Graph of the two real branches of Lambert $W_{k=0,1}$ function and geometrical interpretation based on our simple cliff-shoreface model assuming $(c_0/c_s) = 1$ or that cliff and shoreface sediment composition are the same. The black arrows indicate the direction towards which an initial state will evolve over time. Points I, II and III are all on the principal branch $W_{k=0}$, and all tend toward equilibrium (x = 0), while Point IV is the only one over $W_{k=1}$ and represents an unstable point that will evolve towards -∞ over time.

Equipped with Equations (20)–(22), we can now infer if the shoreline is likely to respond to a sea-level rise by eroding or advancing and at what rate of change (i.e., maximum, medium or minimum).

## 2.2. Characterization of Great Britain-and-Isles Cross-Shore Profiles: Overview

We followed a five-step methodology to extract and classify the elevation transects for the GB coastline as illustrated in Figure 6. Steps 1 to 3 extracted the elevation profiles in a traceable and repeatable way, while Steps 4 and 5 clustered the extracted profiles into types of elevation transects that could be linked with the anticipated geomorphological change due to a sea-level rise. The number of elevation transects will depend on the resolution of the Digital Elevation Model (DEM) and the coastline length and location, and the actual elevation transect geometry will depend on how the orthogonal transects are defined. The method chosen will need to be able to effectively handle a large number

(in the order of millions of transects) to cover the extent of the whole coastline of GB, including the islands, which is 31,368 km, according to Ordnance Survey (OS). To delineate the orthogonal transects, we chose the same approach used in the CliffMetric algorithm proposed by [9], which was specifically designed to resolve coastlines with very irregular shapes such as the GB coastline. As our goal was to assess the likely response of the different elevation transects to a sea-level rise, based uniquely on backshore topographical data, we selected the OS High Water Line as the preferred coastline for generating the orthogonal transects. The CliffMetric code proposed by [9] automatically delineates the coastline for a given still water level but does not allow the user to define a coastline as an input. Our first step was then to modify the CliffMetric code to add the option of a user-defined coastline. This improved version of CliffMetric was then used to extract the elevation profiles from a DTM provided by BlueSky International Limited (5 m resolution) and the most up-to-date available DTM for GB (BlueSky is a commercial product subject to license). The dimensionality of the transects extracted was then reduced using Principal Component Analysis (PCA) [11]. The reduced dimension set of the elevation transects was then clustered using the K-means-partitional clustering approach in MATLAB [12]. We chose a partitional clustering approach over a hierarchical approach due to the large number of transects on the order of $10^6$ that we needed to cluster: the number of distance calculations for hierarchical approaches increases geometrically with the number of observations. The partitional approach is an iterative process in which experts have to assess the number of clusters and the algorithm iteratively guesses the centroids or central point in each cluster, and assign points to the cluster of their nearest centroid. The sensitivity of the number of clusters and location was tested by using different number of clusters from 5 to 10. We found that ten clusters were the minimum required to separate all types of environments; in particular, we needed to increase from 9 to 10 to be able to separate the low-lying cliff of East England (Cell 3) from the abundant cluster 8. The sensitivity to the seed centroid was tested by running the clustering ten times for Nc = 10, mapping the clusters over aerial photography as shown in Figure 12 and confirming that the regional statistics remained unchanged (i.e., that the top five dominant clusters per region remained dominant).

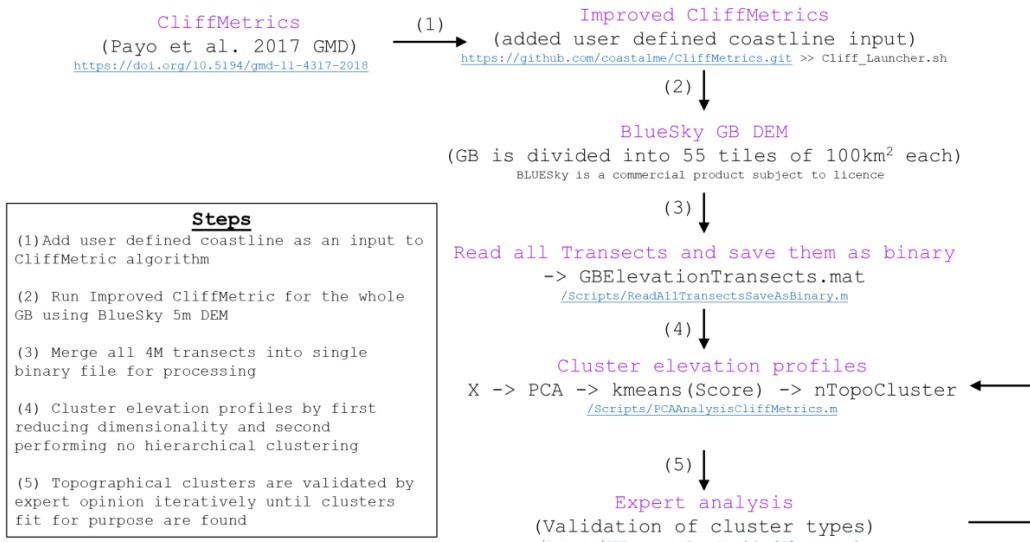

**Figure 6.** Methodological approach followed to classify the elevation profiles along the Great Britain coastline. The scripts used for the different steps are indicated and included as supplementary information.

*2.3. Datasets: BlueSky DTM and HWM Coastline*

To define the morphology of the GB coastline, elevation data were extracted using a DTM provided by BlueSky International Limited (referred to from this point as the BlueSky DTM). This is a 5 m-resolution DTM comprised of both aerial photogrammetry and airborne LiDAR data that covers

GB and the islands. The dataset has a multi-temporal resolution from 2006, with the majority of the data being available post-2015 to 2017. The vertical uncertainty of the dataset is estimated as being ±0.5 m.

The transects used in this study to constrain the backshore topography orthogonal to the coastline were based on the vectorised high water mark (HWM) as extracted from the OS OpenMap Local dataset, version January 2020. In England and Wales, this is the mean level of all the high tides; in Scotland, this is the mean level of the spring high tides. In places where there is no foreshore (e.g., vertical cliffs), the tidal boundary is classified as the high water mark. This dataset has a recommended viewing scale of 1:3000 to 1:20,000. As with the elevation data, these were available as 100 km$^2$ tiles of vector polygon data. To extract the coastline from this dataset, the 100 km$^2$ tiles were dissolved into a single vector dataset. The merged polygon data were then converted into a vector line format, following which the outer extents of all features were removed, leaving a single polyline representation of the coastline. This polyline representation of the data was required, as for this study, the improved CliffMetrics program, which was used to facilitate the data extraction, was extended to take in this externally defined coastline in vector polyline format (see Section 2.4). The manipulation of the OS OpenMap Local data to extract the HWM polyline as described means that we represented the coastline as being the border of polygons available within the OS OpenMap Local dataset. Given the finest scale recommended for viewing this dataset of 1:3000, we estimated that this related to a potential uncertainty of ±1.5 m associated with the position of the HWM line, considering that the Minimum Discernable Mark (MDM) of a 1:3000 scale dataset would be about 1.5 m [13]. The purpose of the coastline transects developed and used in this study was to provide a general characterization of the coast. The vertical (±0.5 m) and temporal uncertainty concerning the temporal resolution of the elevation data regarding their acquisition period (2006–2017), combined with the uncertainty relating to the positioning of the HWM (±1.5 m), are not believed to be of concern and are consequently not considered further in this manuscript.

### 2.4. Improved CliffMetric Algorithm

To extract the elevation profiles, we used an improved version of the CliffMetric algorithm for automatic cliff top and toe delineation [9]. The CliffMetric algorithm builds upon existing methods but is specifically designed to resolve very irregular planform coastlines with many bays and capes, such as parts of the coastline of Great Britain. The original algorithm automatically and sequentially delineates and smooths shoreline vectors, generates orthogonal transects and elevation profiles with a minimum spacing equal to the DEM resolution, and extracts the position and elevation of the cliff top and toe. The outputs include the non-smoothed raster and smoothed vector coastlines, transects orthogonal (hereinafter referred to as transects) to the coastline (as vector shape files), xyz profiles (as comma-separated-value, CSV, files), and the cliff top and toe (as point shape files). The algorithm also automatically assesses the validity of the profile and omits profiles that are too short (i.e., the extraction of the cliff top and toe is not possible when the profile only contains two points). The length of the profiles is initially defined by the user, and the algorithm will attempt to generate the elevation along the full length at every point along the coast. For very irregular coastlines, with many capes and bays, the transects often cross over the coastline before reaching the full (pre-defined) length. For those profiles that cross the shoreline, their length will be reduced to the location where the coastline is crossed. The number of points along the transect will depend on the DEM resolution and orientation of the transect relative to the grid. We used profile lengths of 500 m (i.e., as adopted in the Eurosion project [14]) and a DEM of a 5 m resolution, which means that we had circa 100 points per full-length transect. The elevation of the points along the transects was obtained as the elevation value of the DEM cell whose centroid was closest along the transect. If the algorithm only found two points along a non-full-length normal, the normal was marked as non-valid and no elevation profile was extracted.

We improved the CliffMetric algorithm by allowing the user to define a vector coastline along which the user would like the transect to be generated instead of using the automatically delineated coastline vector. The user needs to enter two additional parameters (Figure S7), which are the shape

file with the user-defined coastline that will be used to generate the orthogonal transects and the sea handiness, indicating on which side of the coastline the sea is: left or right when traversing the points along the coastline. The user-defined coastline needs to be provided in shape file format and point geometry. The algorithm will traverse the coastline, sequentially, from the second feature (i.e., point) in the geometry layer until the second-to-last feature (i.e., ascending ordinal). The orthogonal transect is obtained as the line defined by the centroid of the raster cell closest to the coastline point (i.e., chainage = 0) and the landward point of the transect (i.e., chainage ≤ user defined length + one diagonal cell). The location of the landward point is calculated as the end of a line orthogonal to the local coastline orientation: the coastal orientation is obtained as the straight line linking the coastline points before and after "this" coastline point. The second new input parameter, the sea handiness, is then used to decide on which side of the coastal orientation the landward point of the profile is located. Notice that, opposite to when the coastline is automatically delineated, the improved CliffMetric version uses the user-defined coastline as given without smoothing. We ran the improved CliffMetric algorithm on the BGS-High Performance Computer, which can process the DEM tiles and the outputs. The extracted CSV files with the elevation profiles contain the following columns: "Dist", "X", "Y", "Z" and "detrendZ", which correspond with the chainage in DEM units (e.g., metres); the coordinates X,Y in the DEM coordinate system (i.e., the British National Grid, EPSG: 27700) of the cell centroids from which the elevation, Z, was extracted; and the de-trended elevation, also in DEM units, respectively.

## 2.5. Cluster Analysis of Elevation Profiles

Due to the large dimensionality, (ca. 4 million points x ca. 100 elevation values per transect), some pre-processing was needed to be able to perform a cluster analysis of the extracted transects. The automatic extraction procedure produced a set of transects where most of the transects were of the same length (i.e., 500 m for this work) or shorter (i.e., if the normal was shortened by the algorithm) but did not necessarily have the same number of cross-shore points (i.e., it depended on how many raster cells were underneath each coastline transect). To ensure that all the profiles had the same number of cross-shore points, we normalized the length of all the profiles to the maximum length and resampled at $nx = 100$ cross-shore locations, obtaining a matrix of $nx \times np$, where $np$ is the number of transects used for this analysis. By normalizing the profile length, we are implicitly assuming that the variance of the length is non-informative, which seems reasonable given the transect length is a user pre-defined variable. We used the de-trended elevation at each of the $nx$ cross-shore locations to ensure that the elevation at the seaward limit of the transect ($x/x_{max} = 0$) and landward limit of the transect ($x/x_{max} = 1$) was the same: this ensured that the data were periodic and easier to fit to a set of orthogonal Eigen functions.

The matrix dimension was reduced to a matrix of $nc \times np$, where $nc \leq nx$ is the number of main Principal Components (PCs) obtained from applying the Principal Component Analysis [15] that capture a given percentage of the total database variance. Principal Component Analysis (PCA) is a technique for reducing the dimensions of large datasets, increasing interpretability but at the same time minimizing information loss. The empirically generated PCs have a series of properties that make them suitable for reducing the dimensionality of a dataset: (1) they are a reduced set of variables that are optimal predictors for the original variables in the sense of the square minimums, and (2) the new variables are obtained as an original combination of the former ones and are not correlated. By reducing the dimensions of the database using the PCs, we avoided including some of the non-informative small profile differences in the cluster analysis. The matrix algebra required is standard on commercially available routines (e.g., the pca function in MATLAB); thus, it is not discussed in more detail here. For the interested reader, the MATLAB script used is provided as supplementary material.

A detailed description of cluster analysis is beyond the scope of this paper, and the reader is referred to Hennig, et al. [16] for a general description and to [17,18] for updated references to coastal applications. In brief, it entails the calculation of distances (interpreted as the similarity) between all objects in a data matrix, on the basis that those closer together are more alike than those are

further apart. The clustering techniques can be generally classified as hierarchical or partitional approaches [12]. Hierarchical, agglomerative clustering is a bottom-up approach where objects and then clusters of objects are progressively combined on the basis of a linkage algorithm (or dendogram) that uses the distance measures to determine the proximity of objects and then clusters to each other. The number of distance calculations in hierarchical approaches increases geometrically with the number of observations, making this method unsuitable for very large datasets. Partitional methods have advantages in applications involving large datasets for which the construction of a dendogram is computationally prohibitive. A problem accompanying the use of a partitional algorithm is the choice of the number of desired output clusters. The partitional technique usually produces clusters by optimizing a criterion function defined either locally or globally. A combinatorial search of the set of possible labelling for an optimum value of a criterion is clearly computationally prohibitive. In practice, therefore, the algorithm is typically run multiple times with different starting states, and the best configuration obtained from all of the runs is issued as the output clustering. One such partitional technique is the K-means approach [12]. K-means is one of the simplest unsupervised learning algorithms that solves the well-known clustering problem. The procedure is a simple and easy way to classify a given dataset through a certain number of clusters (assume *k* clusters) fixed a priori. The main idea is to define *k* centroids, one for each cluster. The next step is to take each point belonging to a given dataset and associate it to the nearest centroid. When no point is pending, the first step is completed and an early group is performed. At this point, it is necessary to re-calculate *k* new centroids as centres of the clusters resulting from the previous step. After these *k* new centroids, a new binding has to be performed between the same data points and the nearest new centroid. A loop is generated. As a result of this loop, it may be noticed that the *k* centroids change their location step by step until no more changes are made. In other words, centroids do not move any more. We used the "Manhattan distance metric" as the distance metric because this metric provides more meaningful results from both the theoretical and empirical perspectives for large datasets [19].

## 3. Results

### *3.1. CliffMetrics and Principal Component Analysis Outputs*

Out of the 4,405,501 points from which the user-defined coastline for GB is made, we obtained 4,289,708 profiles. There are 115,793 profiles fewer (2.6%) than the number of coastal points and profiles. The majority (97%) of the profiles or 4,153,505 were considered valid, and only 136,203 profiles (3%) were non-valid because they were too short to be included in the analysis: a profile was considered too short if the length was no longer than one diagonal cell or 7.07 m for the 5 × 5 m cell size used in this study. As expected, the median length of the valid profiles was 500 m, with a minimum length of 10 m and, unexpectedly, a maximum length of 2144 m: we found that 52 profiles were longer than 508 m. The minimum expected elevation for the BlueSky DTM was −5 m, with no data being defined as −3.402e+38. For the profile cluster analysis, we ensured that we used only profiles that passed the quality check and did not include flat profiles or elevation profiles with data gaps. The total number of valid transects was reduced to 3,912,935 after we eliminated those profiles whose minimum elevation and elevation at the landward end were larger than 0.1 and −5 m, respectively, keeping the transects of length smaller than or equal to the user-defined length (508 m for our study).

The majority (99%) of the variability observed in the de-trended elevation profiles is explained by a reduced set of ten PCs. Figure 7a shows the cumulative percentage explained by the first 20 PCs obtained by applying the PCA to all 3,912,935 valid profiles. The cumulative percentage of variability explained increases rapidly with the first few PCs, with the first two PCAs alone explaining 86% of the total observed variability. The first 10 PCs explain 99% of the total variability. The mean differences between the original elevation profile and the re-constructed elevation profile using the first 10 PCs are less than ±15cm, as shown for the profile in Figure 7b. Using the first 10 PCs allows us to reduce the

dimensions of the observation matrix by a factor of 10, from 100 observations for each transect to just 10 PCs, while explaining 99% of the total variability.

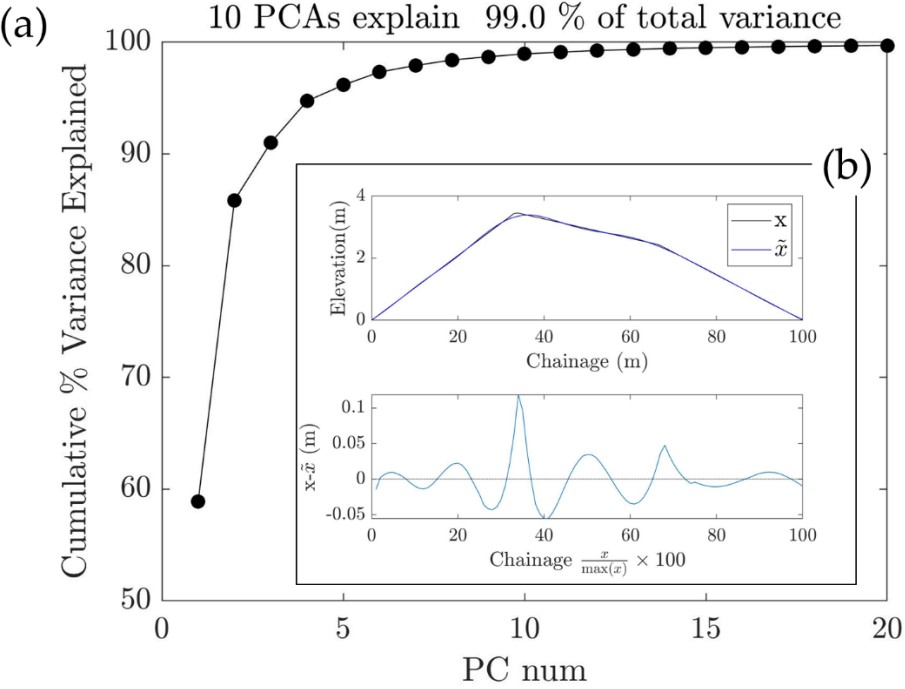

**Figure 7.** The Principal Component Analysis (PCA) suggests that we can capture the majority of the variability observed on the de-trended elevation profiles with a reduced number of Principal Components (PCs): (**a**) percentage of variability explained by the first 20 PCs. The percent of cumulative variance explained is on the y-axis, and the Principal Component (PC) number is on the x-axis; (**b**) comparison of one of the original elevation profiles, x, versus the reconstructed, $\widetilde{x}$, using the first 10 PCs. Top panel shows the elevation profile, and bottom panel, the absolute difference between the original and reconstructed.

Figure 8 shows the cross-shore eigenvector values for the first three PCs. Note that the first eigenvector of the three modes accounts for the 59% of the total variability. This result is not surprising since we used the non-normalized de-trended elevation profiles, and, consequently, the centroid, defined as the mean of the variables, can explain most of the data. For that reason, the first eigenvectors are often highly correlated with the mean vectors. Thus, the combination of the first eigenvectors gives a representation of the mean de-trended elevation profile. The convex parabolic shape of the mean de-trended profile suggests that a large percentage of profiles do not increase in elevation linearly from the seaside landwards but increase at different rates. The location of the maximum of the de-trended profile indicates the cliff top location. The second and the third PCs explain 27% and 5% of the total variability each. These two PCs can be understood as the deviation from the mean de-trended profile represented by the first PC. The first three PCs explain 91% of the total variability. The closer the score is to zero for each one of the PCs, the closer to a flat profile the de-trended elevation profile is, which means that the elevation increases linearly from the seaside landwards.

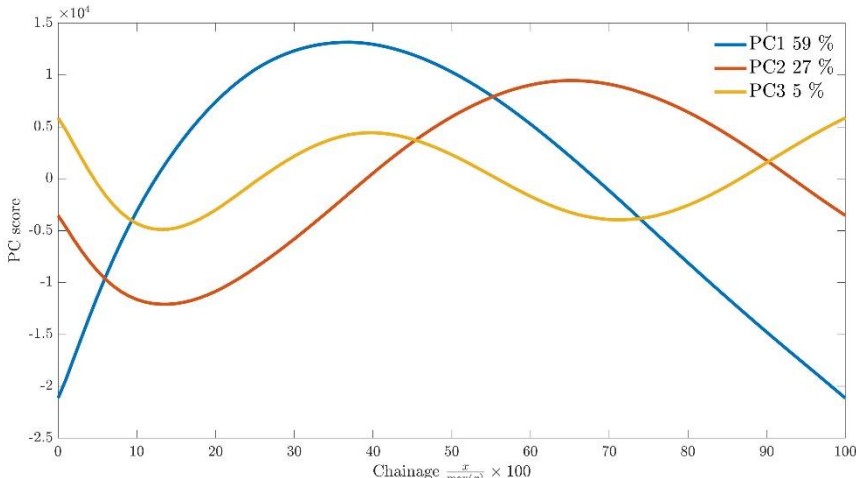

**Figure 8.** Cross-shore eigenvectors for the first three PCs. The percentage shown in the legend indicates the percentage of variability explained by each PC.

Figure 9 shows the centroid and variability of each one of the ten different clusters obtained when plotted against the first and second PCs. Cluster #8 is the most abundant type of profile, representing 48% of all the profiles, while the remaining clusters combined represent the remaining 42% of the profiles. The spreading of Cluster #8 is also significantly smaller that the spreading of the other clusters, as shown by the horizontal and vertical lines that represent twice the standard deviation for each cluster. The low scores for Cluster #8, for both PCs, suggest that profiles belonging to this cluster are close to elevation profiles with constant slopes and low elevation. The large spreading of both PC scores for the other clusters suggests that there is a significant variability in both the cliff top elevation and horizontal location.

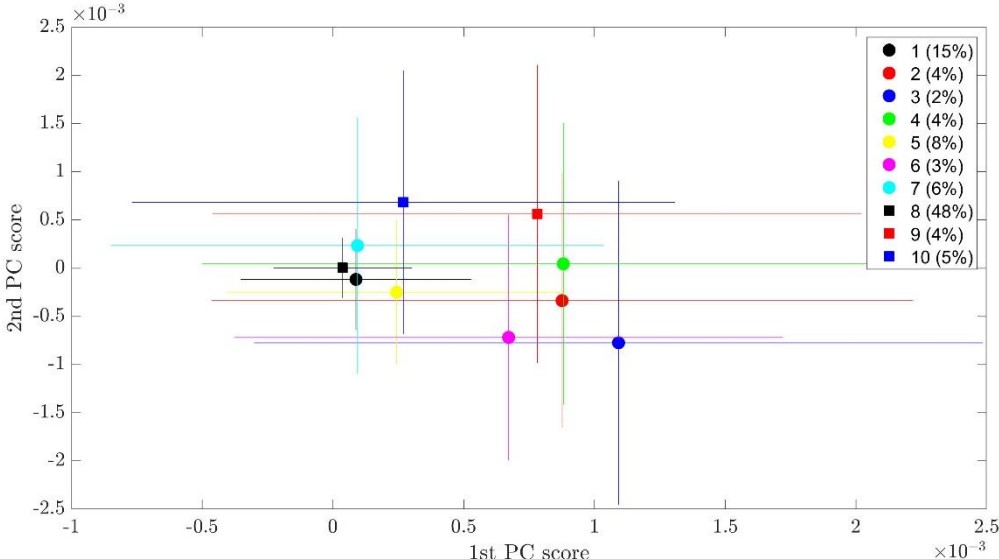

**Figure 9.** Centroids and spreading of scores for all the six clusters obtained. The coloured circle represents the centroid of the cluster, and the solid vertical and horizontal lines represent twice the standard deviation of each PC. The percentage of profiles belonging to each cluster is shown in parentheses in the legend together with the cluster number.

Figure 10a shows the distribution of the maximum profile elevation per cluster as a violin plot. As expected from the PCA results, the maximum elevation of the Cluster #8 profiles is the lowest of all the clusters, with a modal maximum-profile-elevation value of 1 m. The frequency distribution per

cluster shown in the violin plots suggests that the most frequent (mode) and the average (mean and median) profiles are not the same. The most frequent profile elevation is consistently smaller than the median and mean values for all clusters. The differences between the most frequent and averaged elevation profiles is shown in Figure 10b,c, which illustrate the closest-to-centroid and most-frequent elevation profiles for each cluster. The elevation profile for Cluster #8 is the flattest of all the de-trended elevation profiles, irrespectively of if showing the closest-to-centroid (Figure 10b) or most-frequent (Figure 10c) profile. The elevation for the closest-to-centroid profiles for all clusters, but Cluster #8, presents a maximum elevation between the coast point (relative chainage = 0%) and the most landward point of the profile (relative chainage = 100%) (Figure 10b). The elevation profile for Clusters # 1, 2, 3, 4, 5, 6, 7, 9 and 10 differs by both the location of the maximum and the elevation of the maximum. In chainage relative units, the location of the maximum of the de-trended elevation profiles for each cluster varies from 5% for Cluster #1 to 85% for Cluster #7.

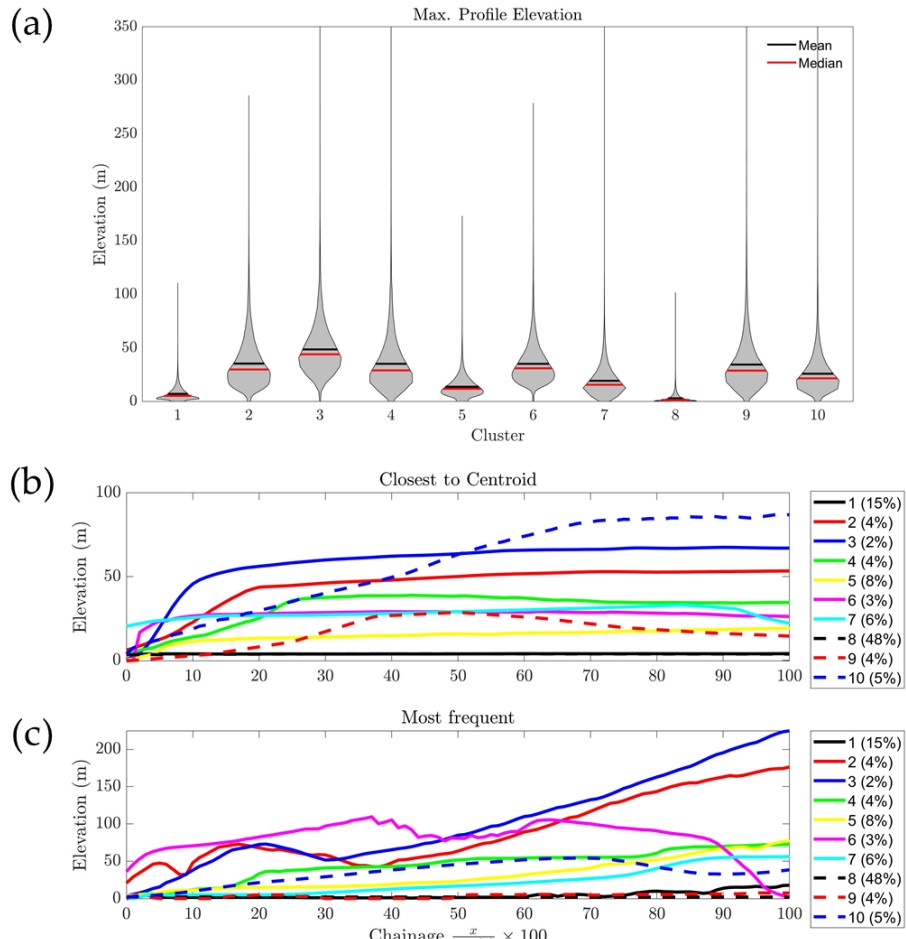

**Figure 10.** Statistical comparison between clusters: (**a**) violin plot of maximum elevation per cluster; (**b,c**) elevation profiles for each cluster showing the profile closest to the centroid (**b**) and most frequent profile (**c**). The percentage of profiles belonging to each cluster is shown in parentheses in the legend together with the cluster number.

Table 1 and Table S2 show the elevation and de-trended elevation profiles for each cluster for a representative profile per cluster. The representative profile was chosen as either the one closest to the cluster centroid (Figure S2) or the one closest to the most frequent value (Figure S3). Despite this variability, when the clusters are mapped over an aerial view of the backshore topography (Figure 11), it seems clear that Clusters 1, 5 and 8 correspond with either gentle profiles or back-barrier-type profiles. Examples of back-barrier beach along the GB coastline can be found at Chesil beach and Blakeney,

which was classified mostly as Clusters 8, 1 and, to a lesser degree, 5. The variations of the cluster type observed at the Blakeney area seem to be related with the relief at the end of the shoreline normal. The Wash is a very low-lying area that was also dominantly classified as Clusters 8 and 1. The high cliffs around St. Bees and Flamborough head were mostly classified as Clusters 2, 3 and, mostly, 6. No clear pattern in the spatial distribution was observed for Clusters 4, 7, 9 and 10. The spatial distribution of the topographical clusters is best understood by plotting the frequency distribution per region.

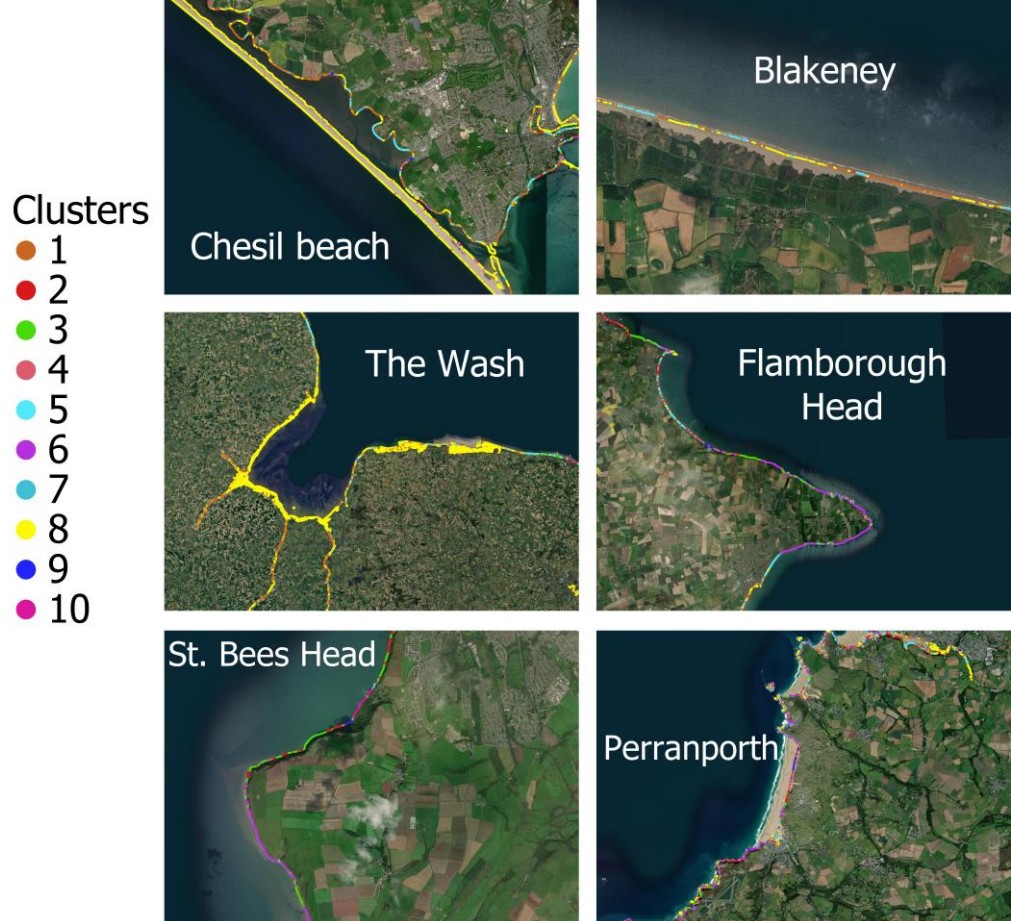

**Figure 11.** Cluster types mapped to the coast point along the high water mark line used to delineate the transects. Names of the locations around the GB coastline indicated as text. Source of aerial imagery: Esri, Maxar, GeoEye, Earthstar Geographics, CNES/Airbus DS, USDA, USGS, AeroGRID, IGN, and the GIS User Community, v10.6.

We used the eleven sediment cells defined by the Shoreline Management Plans (second revision) for England and Wales and the eleven administrative units of Scottish marine regions (see Figure 12) to extract regional statistics. The Scottish Marine Regions are 11 areas established for the purpose of regional marine planning, defined by The Scottish Marine Regions (SMR) Order 2015. These SMR regions are sub-areas of both the "Scottish marine area" defined in the Marine Act 2010 and "Scottish inshore region" defined in the Marine and Coastal Access Act 2009. From the regional counts on cluster types (Supplementary Materials), it is clear that Cluster 8 is the most abundant cluster type for all 22 regions (Figure 12). Clusters 3, 6 and 7 are not present on all regions.

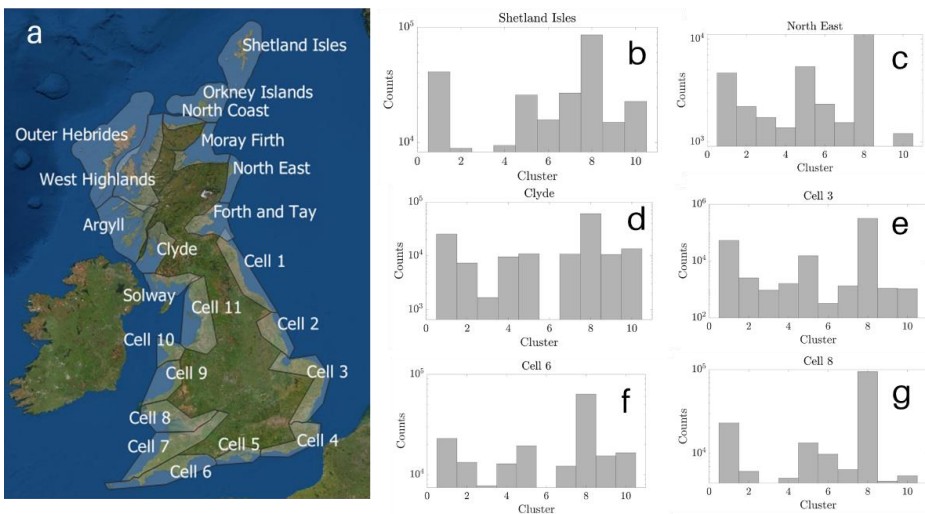

**Figure 12.** Example of the regional statistics that we extracted for the 22 regions into which we divided the GB coastline: (**a**) shows the region coverage; (**b**–**g**) show the counts of topographical cluster types for some regions (all other regions are shown in Supplementary Materials). Source of aerial imagery: Esri, Maxar, GeoEye, Earthstar Geographics, CNES/Airbus DS, USDA, USGS, AeroGRID, IGN, and the GIS User Community, v10.6.

### 3.2. Main Topographical Characteristics

We extracted the geometrical attributes of the backshore topographical profile as shown in Figure 13 for all transects and calculated the regional statistics. The backshore shape parameter, $\alpha_0$, defined for the domain $-\infty < \alpha_0 \leq 1$ represents how much the area under the backshore elevation profile from the coast to the end of the transect differs from a rectangle that has the same length and an elevation equal to the maximum profile elevation, $z_{max}$. Note that we used $z_{max}$ instead of the elevation at the end of the profile, $z_{end}$, because as we saw from the cluster type 7 (Figure 10), $z_{end} \leq z_{max}$, as the elevation does not grow continuously backshore. This shape parameter will be close to 1 if the shape is similar to a rectangle and smaller if it is not. Negative values of $\alpha_0$ can occur for profiles where the elevation is not always larger than or equal to the elevation at the initial point ($x = 0$). We also extracted the backshore slope, $S_0$; cliff face slope, $S_c$; and initial cliff relief, $H_{c,0}$. Figure 14 shows the regional statistics for the clusters interpreted as cliff-type (i.e., Clusters 2 to 4, 6, 7, 9 and 10). We observed that there was a small percentage, of 4%, of the data that produced negative initial cliff relief, suggesting that they did not represent a cliff coast either, and they were omitted for our assessment of the cliff coast response to a rising sea level around the GB coastline.

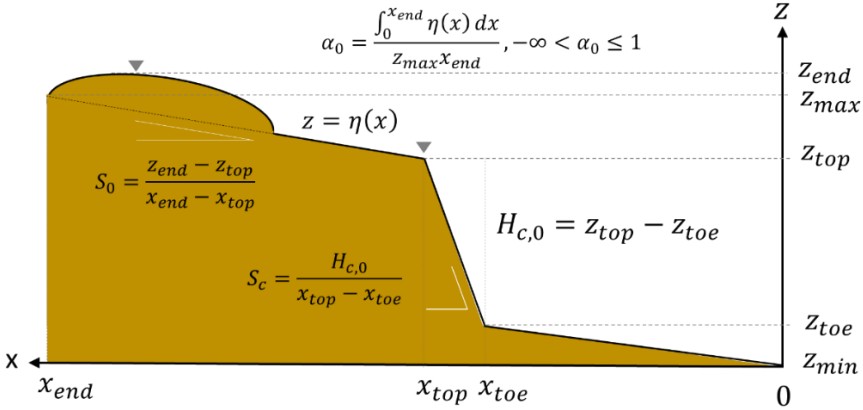

**Figure 13.** Main geometrical attributes extracted from the elevation profiles along the GB coastline.

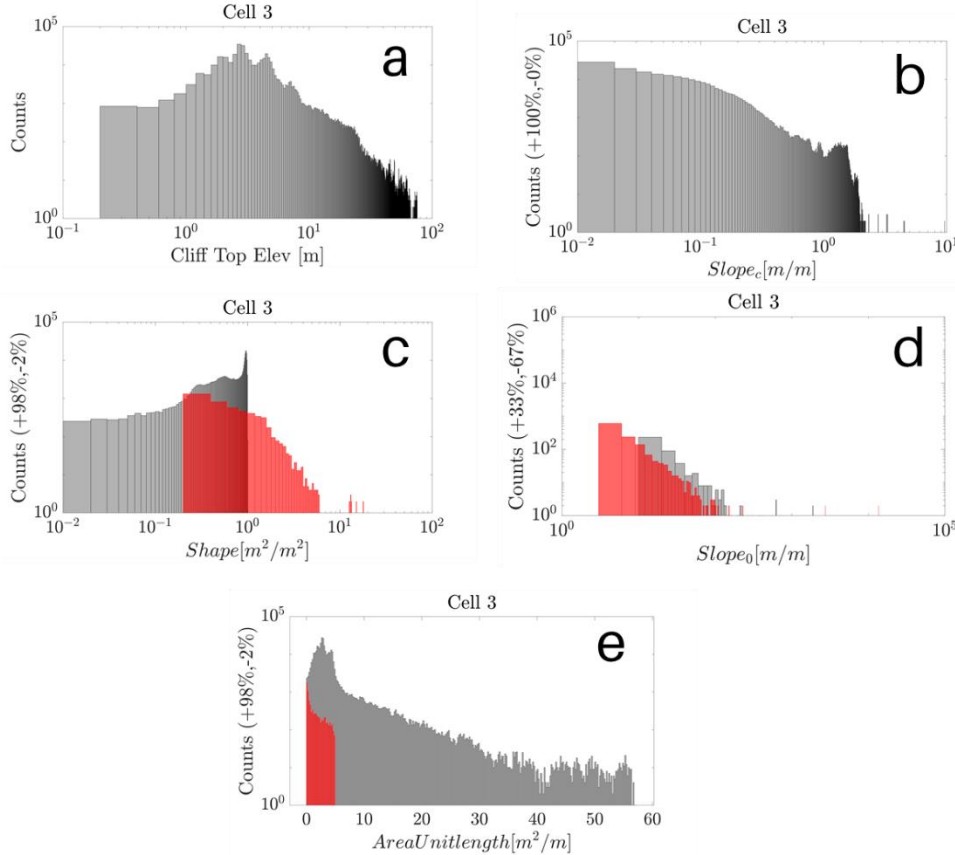

**Figure 14.** Example of the regional statistics that we extracted for the 22 regions into which we divided the GB coastline. As an example, we show here the stats for region "Cell 3", and all other regions are in Supplementary Materials; panels show the counts per region of the cliff top elevation (**a**), slope of cliff face (**b**), shape of the inland topography (**c**), inland slope (**d**) and area per unit length (**e**). These geometrical attributes were calculated as shown in Figure 14. Region area is shown in Figure 13.

Figures 15 and 16 show the regional statistics for the initial cliff relief $H_{c,0}$ and backshore slope, $S_0$, for all the interpreted cliff profiles. To obtain these stats, we did not include clusters that corresponded with non-cliff coastlines (Clusters 1, 5 and 8). We observe that the majority (57%) of the backshore slope values are negative and 43% have positive slopes. The modal, median and standard deviation values for the positive and negative slopes are of the order of $(3.3 \times 10^{-2},\, 9.6 \times 10^{-4}$ and $4.1 \times 10^{3})$ and $(-4.8 \times 10^{-4},\, -2.9 \times 10^{-2}$ and $-3.4 \times 10^{3})$, respectively. As we show in Figure 5 and Equation (9), profiles with negative backshore slopes will not evolve towards an equilibrium cliff relief but will tend to decrease towards $-\infty$ at the fastest rate of change.

Figure 17 shows the $z_c$ for those transects interpreted as cliff types and whose backshore slopes are positive and therefore will evolve towards an equilibrium relief. We calculated the $z_c$ values using Equation (20) and assuming the values of the shoreface length, $L_s = 10^3$ m, and shoreface relief, $H_s = 10^1$ m, as representative values for the GB coastline. All the $z_c$ values obtained are positive, with a mode, median and standard deviation equal to 13.9, 1.1 and $2.5 \times 10^6$, respectively. The modal value being of the order of $10^1$ suggests that the most likely state of the profiles is to be not in equilibrium yet. The large values also indicate that the rate of change is smaller than the one that will be realised as the cliff relief gets closer to the equilibrium relief. The median value being close to 1 indicates that half of the locations are close to equilibrium and in the region where the rate of change is mostly determined by the backshore slope.

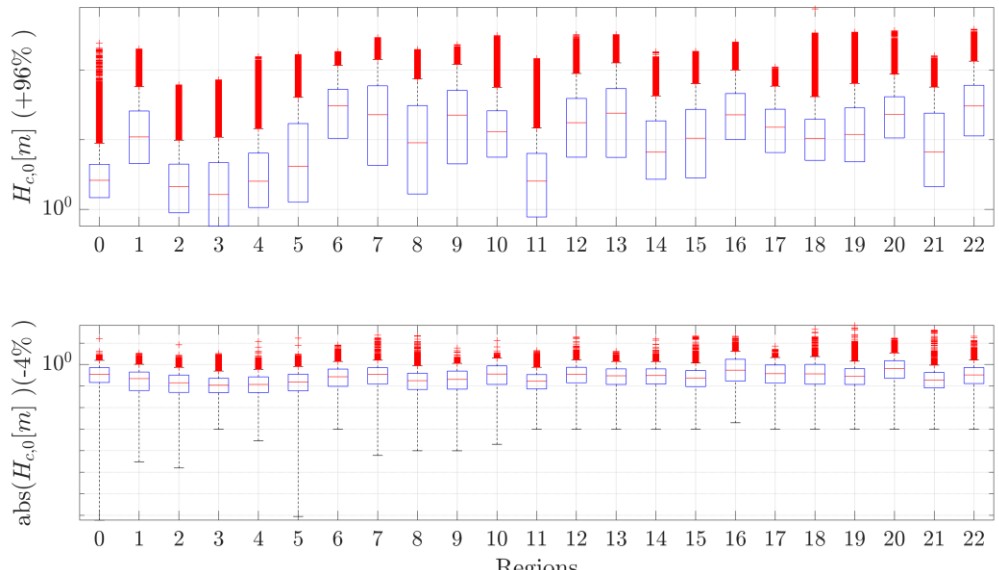

**Figure 15.** Regional statistics of initial cliff relief $H_{c,0}$ along GB coastline. The central mark indicates the median, and the bottom and top edges of the box indicate the 25th and 75th percentiles, respectively. The whiskers extend to the most extreme data points not considered outliers, and the outliers are plotted individually using the "+" symbol. See Table S1 for correspondence between region number and admin unit used.

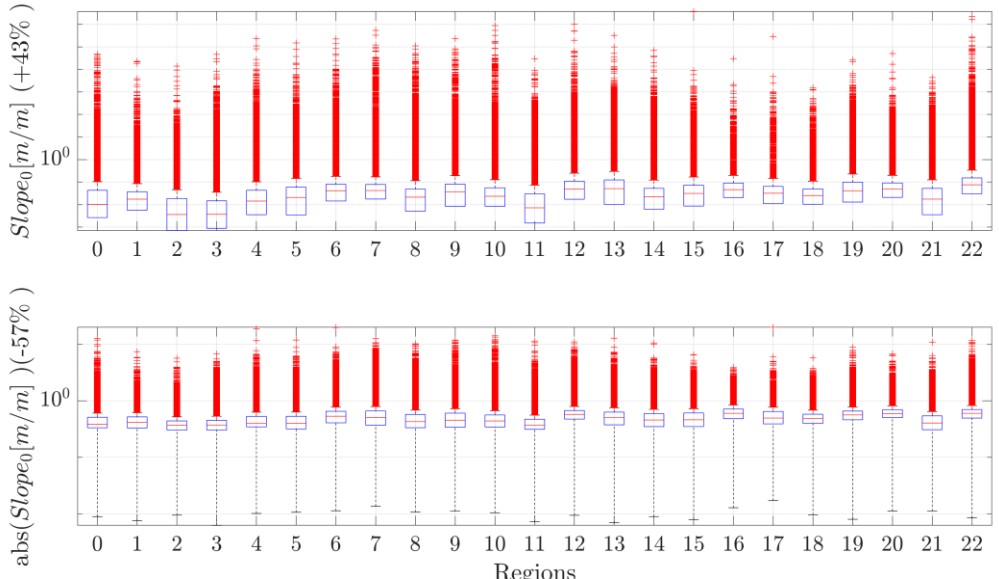

**Figure 16.** Regional statistics of initial backshore slope $S_0$ along GB coastline. The central mark indicates the median, and the bottom and top edges of the box indicate the 25th and 75th percentiles, respectively. The whiskers extend to the most extreme data points not considered outliers, and the outliers are plotted individually using the "+" symbol. To obtain these stats, we did not include clusters that corresponded with non-cliff coastlines (Clusters 1, 5 and 8) and 4% of other clusters that had $H_{c,0} < 0$. See Table S1 for correspondence between region number and admin unit used.

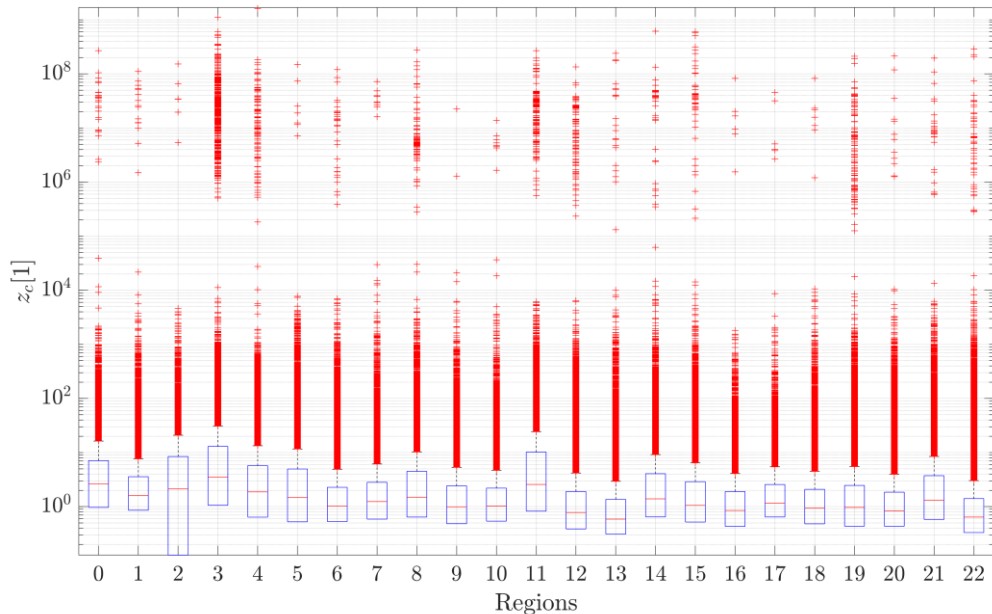

**Figure 17.** Regional statistics of $z_c$ along GB coastline. The central mark indicates the median, and the bottom and top edges of the box indicate the 25th and 75th percentiles, respectively. The whiskers extend to the most extreme data points not considered outliers, and the outliers are plotted individually using the "+2" symbol. To obtain these stats, we did not include clusters that corresponded with non-cliff coastlines (Clusters 1, 5 and 8) and 4% of other clusters that had $H_{c,0} < 0$ and $S_0 \geq 0$. We used Equation (20) and assumed a shoreface slope representative of GB's open coast to be equal to $10^{-3}$. See Table S1 for correspondence between region number and admin unit used.

## 4. Discussion

### 4.1. Interpretation of the Different Clusters

Each cluster can be described using a number of parameters including the shorezone width, inclination of the backshore slope, minimum and maximum elevations of the profiles, profile curvature and distance from the high-water mark to cliff top. These are parameters that we extracted for all profiles and analysed based on the mean and range of each parameter for each cluster. Below, we summarise the characteristics of each cluster, the broad spatial extent and frequency of occurrence of each cluster, and the dominant geology or coastal environments associated with each cluster. Figures S1 and S2 show the closest-to-centre-point and the most-frequent profiles for each cluster and are helpful in visualising the geometry of the clusters.

*Cluster 1—Low-Coastline Back-Barrier Type (Shallow Angle)*

Generally characterized by a narrow shore, a gently inclined slope that reaches a very low elevation a short distance from the HWM, a very low profile curvature and low maximum elevations. This cluster accounts for 15% of the UK coastline and is typically well distributed around the coastline. It is more prevalent in areas such as eastern England and the north coast of Wales. It is commonly found in estuarine regions, where soft superficial deposits dominate the geology.

*Cluster 2—High Cliff Coastline*

Generally characterized by a minimal shore, steep and high cliffs, a moderate distance from the HWM to the cliff top, a moderate profile curvature and high maximum elevations. This cluster accounts for 4% of the UK coastline and is more common on the west coasts of England, Wales and Scotland. These parts of the coastline are typically dominated by hard, strong bedrock.

*Cluster 3—High, Sheer Cliff Coastline*

Generally characterized by a minimal shore, very steep and high cliffs, a short distance from the HWM to the cliff top, a very high profile curvature and high maximum elevations. This cluster accounts for 2% of the UK coastline and is well distributed around the coast but is more common in the southwest and around northern Scotland. Typically, these parts of the coastline are dominated by hard, strong bedrock.

*Cluster 4—Narrow Beach and Cliff Coastline*

Generally characterized by a narrow shore, steep and high cliffs, a moderate distance from the HWM to the cliff top, a moderate profile curvature and high maximum elevations. This cluster accounts for 4% of the UK coastline and is found more around southwest England and the west coast of Scotland. These parts of the coastline are dominated by hard, strong bedrock.

*Cluster 5—Low Coastline Back-Barrier Type (Moderate Angle)*

Generally characterized by a narrow shore, a moderately steep and low cliff, a short distance from the cliff top to the HWM, a low profile curvature and low maximum elevations. This cluster accounts for 8% of the UK coastline and is well distributed around the coastline, although it is more common along the coast of east and northeast England. These coastlines are commonly straight and dominated by weak, poorly consolidated superficial deposits.

*Cluster 6—Plunging Coastline*

Generally characterized by a minimal shore, steep and moderately high cliffs, a very short distance from the HWM to the cliff top, a very high profile curvature and moderate maximum elevations. This cluster accounts for 3% of the UK coastline and is most common in the Pembrokeshire region of west Wales and along the NW coast of Cornwall. These parts of the coastline are typically formed of hard, strong, consolidated bedrock that is highly resistant to erosion.

*Cluster 7—Wide Shore Coastline*

Generally characterized by a wide shore, a steep and moderately high cliff, a long distance from the HWM to the cliff top, a low profile curvature and moderate maximum elevations. This cluster accounts for 6% of the UK coastline and is mostly found along the west coast of Scotland and around the Scottish isles. The coastline in these areas is typically formed of hard, strong (commonly metamorphic) bedrock, which often forms wide wave-cut platforms and complex shaped inlets.

*Cluster 8—Flat Coastline*

Generally characterized by a narrow shore, a flat or very gently rising slope, a short distance from the HWM to the top of the slope, a very low profile curvature and very low maximum elevations. This cluster accounts for 48% of the UK coastline, being by far the most prevalent cluster, and is predominantly found around estuaries, such as the Medway and Humber.

*Cluster 9—Retrograde Slope Coastline*

Generally characterized by a narrow shore, a steep and moderately high cliff, a long distance from the HWM to the cliff top, a low profile curvature, high maximum elevations and a retrograde slope after maximum elevation. This cluster accounts for 4% of the UK coastline and is most common along the west coast of Scotland and along the south coast of Cornwall. These regions of the coastline are commonly dominated by hard, strong bedrock.

*Cluster 10—High Rising Coastline*

Generally characterized by a wide shore, a steep and moderately high slope, a long distance from the HWM to the cliff top, a low profile curvature and moderate maximum elevations. This cluster accounts for 5% of the UK coastline and is most common along the west coast of Scotland and along the south coast of Cornwall. These regions of the coast are commonly dominated by hard, strong bedrock.

*4.2. Likely Response to Sea-Level Rise of GB Coastline*

Our cluster analysis (Figure 10) and interpretation (Figure 11 and Section 4.1) suggest that 71% of the coast of GB is best described as gentle coast, including estuarine coastline or open coasts where back-barrier beaches can form. The remaining 29% is best described as cliff-type coastlines. We have shown how using the geometrical relationships obtained from applying sediment conservation over a control volume that includes the backshore topography and sand sediment fraction (i.e., Equations (9) and (12)), we can infer the likely response of any gentle and steep profile to a sea-level rise. In particular, we have shown how we can qualitatively assess the rate of change (Figure 5) and the net behaviour of the shoreline (i.e., erosion or accretion) by assessing the disequilibrium state of the backshore profile (Equation (22)). We observed that the majority (57%) of the backshore slope values (as defined in Figure 13) are negative and 43% positive. This observation suggests that a large proportion of GB's coastline will respond by very rapidly retreating, with insufficient local source material to counteract the rapid regression, and will therefore require non-local sources of sand and other beach materials to maintain the actual shoreline position as the sea level rises. The number of potential sources of sand can be inferred by assessing the condition of the shoreline accretion of cliffed coast from Equation (22) as shown in Figure 18. If the initial cliff relief, $H_{c,0}$, is larger than the equilibrium cliff relief, $H_{c,\infty}$, the necessary but not sufficient condition for shoreline advance from Equation (7) is met: the section will act as a sediment source, and if the net gained area is larger than the area lost due to a sea-level rise, then the shoreline will advance. On the contrary, if $H_{c,0} < H_{c,\infty}$, the only possible shoreline response is to retreat and the coast section will act as a sediment sink. The percentages of sections acting as potential sources of sand are the highest (80% and 73%) in Cell 3 and Cell 11, respectively, and the percentages of sections acting as potential sand sinks are the highest (67% and 66%) in the Clyde and West Highlands. The amount of sediment released per unit of cross-shore length also varies across regions (Figure S4), with median and standard deviation values equal to 11 and 23.3 m$^2$/m, respectively. A first-order estimate of the sand required by a sea-level rise R, assuming that the cliff and shoreface have similar sand fractions, $c_0 = c_s$, can be obtained from Equation (2), as $RxL_s$, where $L_s$ is the shoreface length. Assuming $L_s = 10^3$ metres and R = 0.5 metres, it will demand $5 \times 10^2$ m$^2$/m, which will be equivalent to ca. 500 m/11 m$^2$/m $\cong$ 50 m retreat. How this material is released will depend, among other things, on the shape of the backshore profile, which we found (Figure S5) to vary across regions, with a median and standard deviation of 0.69 and 0.16, with a percentage of less than 1% being negative.

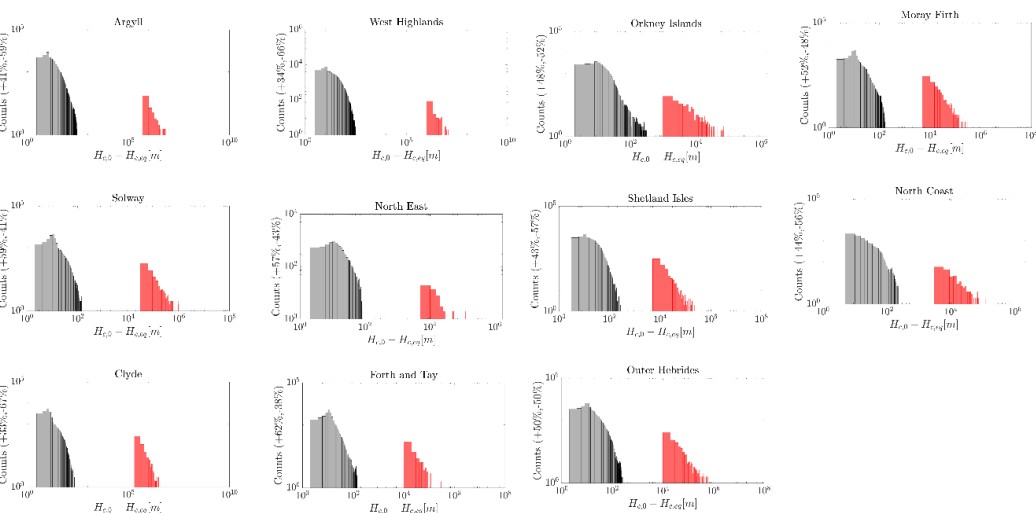

**Figure 18.** *Cont.*

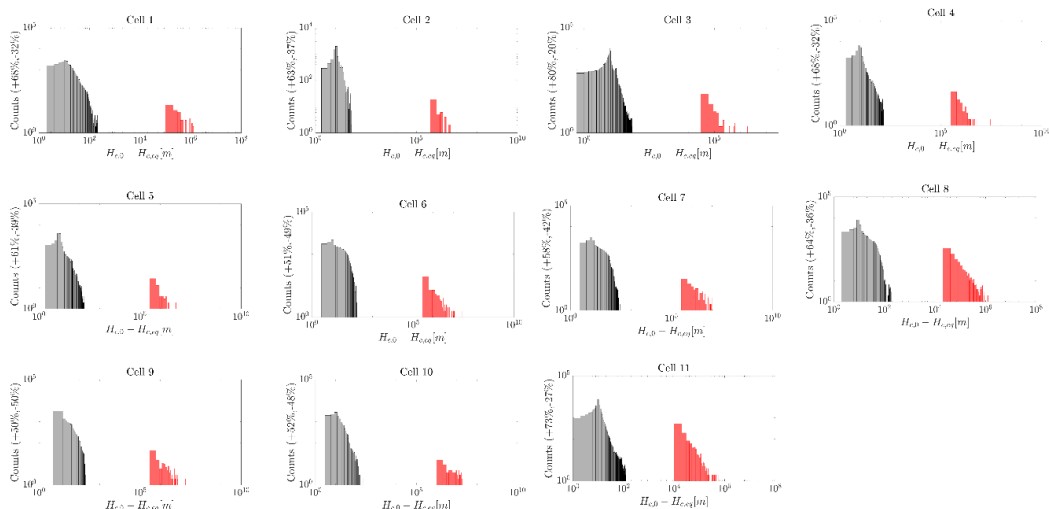

**Figure 18.** Regional statistics of $H_{c,0} - H_{c,\infty}$ along GB coastline. To obtain these stats, we did not include clusters that corresponded with non-cliff coastlines (Clusters 1, 5 and 8) and 4% of other clusters that had $H_{c,0} < 0$ and $S_0 \geq 0$. We used Equation (8b) and assumed a shoreface slope representative of GB's open coast to be equal to $10^{-2}$. Grey bins represent section counts where $H_{c,0} - H_{c,\infty} > 0$ (i.e., potential sediment sources), and red bins represent section counts for which $H_{c,0} - H_{c,\infty} < 0$ (i.e., potential sediment sinks).

### 4.3. Limitations of This Study

The results presented in this work need to be interpreted with caution and an awareness of the limitations of our approach. The number of elevation transects is very sensitive to the DEM resolution and the vector coastline used to delineate the orthogonal transects. The coastline of GB, including the islands, is 31,368 km, according to Ordnance Survey (OS), with the mainland making up 17,819 km. Other institutions have the figure lower—the CIA Fact book says 12,429 km, and the World Resources Institute says 19,717 km. We used a total of 3,912,935 valid transects, which represented ca. 19,565 km and were within the expected length based on other sources. The differences in the coastline length estimations are due to the coastline paradox [20]. The coastline paradox states that a coastline does not have a well-defined length. The measurements of the length of a coastline behave like a fractal, being different at different scale intervals (the distances between points on the coastline at which measurements are taken). The smaller the scale interval (meaning the more detailed the measurement), the longer the coastline. This "magnifying" effect is greater for convoluted coastlines such as the GB coastline than for relatively smooth ones.

Our simplified framework provides valuable insights into coastal evolution, and has the particular advantage of providing simple geometrical reasoning to assess long-term shoreline retreat. However, this approach necessarily neglects many issues that could limit its applicability. Our models generalize the Bruun rule but still assume alongshore homogeneity, a fixed shoreface profile and closure depth, and a sandy nearshore. Further discussion of these assumptions is beyond the scope of this paper, but as noted by [7], we also note that the framework of the shoreline Exner Equation (2) is general enough to accommodate alongshore variability, dynamic profiles, shoreface effluxes and patchy sediment cover. Our geometric analysis ignored the likelihood that on steep coasts, cliff retreat may become weathering-limited in the presence of hard crystalline rock, which will reduce the numbers of both potential sinks and sources, as shown in Figure 18. We also assumed $c_0 = c_s$, which in reality will vary with the lithology that has been affected by coastal erosion. From the geometrical relationships, the reader can infer in which direction our assumptions will change as the $c_0/c_s$ ratio increases and decreases.

## 5. Conclusions

Reliable approaches for assessing the likely shoreline change in response to a sea-level rise over time scales from decades to centuries at the national scale are needed worldwide for long-term coastal management and ensuring that the natural and built environment are adaptable to today's and tomorrow's climate. Here, we used a simple geometrical analysis of the backshore topography to assess the likely response of any wave-dominated coastline to a sea-level rise, and we applied it along the entire Great Britain (GB) coastline, which is ca. 17,820 km long. We illustrated how the backshore geometry can be linked to the shoreline response (rate of change and net response: erosion or accretion) to a sea-level rise by using a generalized shoreline Exner equation, which includes the effect of the backshore slope and differences in sediment fractions within the nearshore. Our analysis suggests that 71% of the coast of GB is best described as gentle coast, including estuarine coastline or open coasts where back-barrier beaches can form. The remaining 29% is best described as cliff-type coastlines, for which the majority (57%) of the backshore slope values are negative, suggesting that a non-equilibrium trajectory will most likely be followed as a response to a rise in sea level. For the remaining 43% of the cliffed coast, we have provided regional statistics showing where the potential sinks and sources of sediment are likely to be.

Our findings have implications for both coastal engineers and stakeholders. For coastal engineers, we have illustrated how the relative importance of both the backshore topography and sediment composition differences between the nearshore and backshore can be assessed using widely available topographical information, sediment conservation principles and first order of magnitude analysis. Coastal stakeholders who manage the risk of coastal erosion and adaptation to rising sea levels can now better quantify at a regional level the percentage and location of potential sections that are likely to behave as sediment sinks and sources to better manage the limited amount of sediment available to compensate the area lost due to a sea-level rise.

**Supplementary Materials:** The following are available online at http://www.mdpi.com/2077-1312/8/11/866/s1. CliffMetric.Zip: improved CliffMetric algorithm used for this study.

**Author Contributions:** Conceptualization, A.P., C.W. and K.A.L.; methodology, A.P., C.W. and A.G.H.; software, A.P.; validation, R.V., A.G.H. and J.R.L.; formal analysis, A.L.L.; investigation, A.L.L.; resources, K.A.L.; data curation, A.P., C.W. and A.G.H.; writing—original draft preparation, A.P.; writing—review and editing, J.R.L., R.V. and C.W.; visualization, A.P.; project administration, K.A.L. and R.V.; funding acquisition, K.A.L. All authors have read and agreed to the published version of the manuscript.

**Funding:** This research received no external funding.

**Acknowledgments:** This manuscript was prepared, verified and approved for publication by the British Geological Survey.

**Conflicts of Interest:** The authors declare no conflict of interest. The funders had no role in the design of the study; in the collection, analyses or interpretation of data; in the writing of the manuscript; or in the decision to publish the results.

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
