# Peer review of "Geometrical Analysis of the Inland Topography to Assess the Likely Response of Wave-Dominated Coastline to Sea Level: Application to Great Britain"

_jmse, doi:10.3390/jmse8110866_

Round 1

Reviewer 1 Report

Interesting research with a huge dataset and sound analysis techniques. I have less experience with the Bruun rule, but I think it is an appropriate approach to simplify coastal behaviour with studies on such a large scale.
My notes:
 -Formula 2: Math is sometime hard to read/follow as not all the terms in for example (2) and (9) are explained, are explained much later or are only referenced to another article
-In Corless et al[ref 10] the W function is defined as W(z)e^W(z)=z. I am missing in the article how this is applied in for example formula 9 and 15.
Does it mean that the term between the [] brackets satisfies the W(z)e^W(z)=z condition? Refer earlier to figure 5 to get an idea of the W function.
-I am missing in section 2 some typical dimensions for the observed phenomena. In the paragraph starting at line number 270 it mentions both barrier beaches and barrier island.
Especially barrier island can be quite large and I can imagine that other processes (like aeolian transport) can play an important role there. With different climate effects on sea (sea level rise)
and air (more extreme weather event) I am wondering how these effects are taken into account in the model. And if they should be included.
-Line 353: Several iterations of clustering were judged by expert judgements. I was wondering what the " means. How was this determined?
-Line 372: How is the high water mark determined. Is it based on a tidal model of long term statistics? And which high water does the article mean? Highest astronomical tide, mean high water springs or mean high water?
The referenced website doesn't provide that much information.
-Line 492: The K-means methods requires a prescribed number of clusters. Have you considered the DB-SCAN method which requires less information about the clusters?
-Figure 16-18: I don't understand the graph's. I presume the a red "+" presents one cluster, but what do the square rectangles with the red line mean. Are there also standard deviation displayed with the dotted line. Also why do the different results not overlap?
Additional text is needed to explain the graph's better. Maybe include a legend.
-Line 804: Can you estimate how much land (area and volume) is lost due to the sea level rise associated with the different climate change regimes (ie. average temperature increase of 1,1.5,2... degrees)?
-I was wondering what the source of the sediment characteristics is. I couldn't find a reference to a source.
-Appendix A is not referenced in the text.

Reviewer 2 Report

This manuscript refers to an interesting topic, that is the necessity of quantitative estimates about the shoreline response to sea level variations. To this scope, the authors describe a geometrical approach, based on an automated procedure, which allows to determine the principal geometrical parameters involved in the Brunn rule as improved by Wolinsky and Murray (2009) (their reference [7]).

I suggest a major review, based on my main concerns as here specified:

  • The introduction section should be more general, presenting other similar cases and works. Rather it focuses only on the description of the Bruun rule, as also the continuous citations to references [4] and [7] prove. Other references?
  • All the first part, that is par. 2.1 up to line 290, should be deleted, as it is simply a repetition of what [7] have already published (included Figures, even if re-arranged). It is redundant. It could be enough to write eq. (5), (9) and (15) with their symbols (and referring once to [7]) and start with the authors’ new contribution to the topic. As well, one sketch is fine to show the geometry.
  • Par 2.2, methodology: I am a bit puzzled by the ‘Expert analysis’ module of the approach. Can the authors explain it better? Is the method sensitive to experts’ skills? Is the expert necessary just to ‘guess’ (line 351) the number of clusters? How could we manage this validation if we don’t have experts? Does the whole method fail?

Minor:

  • Abstract: line 25: twice ‘suggest that’
  • Figure 7: better to put it in the appendix
  • Line 433: ‘is then used’
  • Line 441-444: quite trivial, you can delete it
  • Line 518-521: not clear, rephrase
  • Figure 8: caption too long. You generally explain too much in the caption, while an explanation in the paper text is required (same also for Figure 15, 17, 18, 19)
  • Line 550: ‘cero’ ?
  • At the end of par 4.1 it would eb nice to see a figure like Fig. 13 where the principal clusters along the coast are shown (as stripes of different colors, for example)
  • Line 872: ‘that suggest’ delete
  • References: [3] check format
  • in the first part (that in any case I strongly suggest to delete) please check the reference system (in some figures the x axis is opposite), some symbols, typos and punctuation.   

Round 2
